# Full-Time-Scale Fluid-to-Ground Thermal Response of a Borefield with Uniform Fluid Temperature

**Claudia Naldi * and Enzo Zanchini**

Department of Industrial Engineering, Alma Mater Studiorum University of Bologna, Viale Risorgimento 2, 40136 Bologna, Italy; enzo.zanchini@unibo.it
* Correspondence: claudia.naldi2@unibo.it

**Abstract:** The most accurate method for the design and the simulation of a borehole heat exchanger (BHE) field is employing the fluid-to-ground thermal response of the field, namely the mean-fluid-temperature rise produced by a time-constant thermal power supplied to the fluid. Usually, a short-term and a long-term model are applied, with results matched at a selected time instant. In this paper we propose a method to determine the full-time-scale thermal response of a BHE field that employs one numerical model and yields accurate results with a reasonable computation time. Each BHE is modeled as a one-material cylinder with the same radius as the BHE, surrounded by the ground and containing a heat-generating cylindrical surface whose temperature represents that of the fluid. The condition of uniform fluid temperature and time-constant total power supplied to the fluid, necessary for the long-term accuracy, is obtained iteratively, by imposing at the generating surface uniform time-dependent temperatures that converge to the desired condition. A 2 × 2 square BHE field is employed as an example. The method is recommended to obtain the thermal response of a BHE field with uniform fluid temperature, with high accuracy both in the short and in the long term.

**Keywords:** ground-coupled heat pumps; borehole heat exchangers; fluid-to-ground thermal response; full-time-scale; isothermal fluid; numerical method

## 1. Introduction

Ground-Coupled Heat Pumps (GCHPs) are very efficient systems for building heating and cooling, suitable for reducing $CO_2$ emissions. The performance of these systems has been studied both experimentally and by simulation codes [1–5]. GCHPs usually employ vertical ground heat exchangers, often called Borehole Heat Exchangers (BHEs), having a typical length of about 100 m and a typical diameter of about 15 cm.

Indeed, the ground is a porous medium partially saturated by groundwater, and several authors have shown that the groundwater flow can have relevant effects on the performance of a BHE field [6–10]. However, since reliable data on the groundwater flow are usually not available, the ground is still considered as a homogeneous solid in current methods for the design and the dynamic simulation of BHE fields. Most of these methods employ dimensionless thermal response factors, that are often embedded in simulation codes, such as EED [11], GHLEPRO [12], EnergyPlus [13], eQuest [14]. The thermal response of a BHE field is either the surface-averaged temperature rise at the boundary between BHEs and ground, or the mean-fluid-temperature rise, produced by a time-constant thermal power $Q$ supplied to the fluid. The temperature rise $\theta$ is defined as:

$$\theta = T - T_g, \tag{1}$$

where $T$ is temperature and $T_g$ is the undisturbed ground temperature. The mean-fluid-temperature rise, $\theta_f$, is called fluid-to-ground thermal response.

In order to determine the thermal response of a BHE field, each BHE is usually sketched either as a line with a finite length, or as a cylinder with a finite length, that supplies to the ground a uniform and constant thermal power per unit length, $q_l$, equal to the ratio between $Q$ and the total BHE length. The first scheme is called Finite Line-Source (FLS) model, and the second is called Finite Cylindrical-Source (FCS) model. These models allow evaluating the thermal response of a BHE field by the superposition of the effects of the single BHEs. Thus, one determines the temperature rise $\theta(r, \tau)$ produced by a BHE at a distance $r$ from the BHE axis at time $\tau$, averaged along the BHE length, and calculates the temperature rise $\theta_b(\tau)$ of the surface between BHEs and ground as the sum of the contributions of the single BHEs. The mean temperature rise of the fluid within the BHEs is then calculated as:

$$\theta_f(\tau) \ = \ \theta_b(\tau) + q_l R_b, \tag{2}$$

where $R_b$ is the BHE thermal resistance. Analytical solutions of the finite line-source model were developed by Claesson and Eskilson [15] and by Zeng et al. [16], who presented expressions of $\theta(r, \tau)$ in the form of a double integral. The solution has then been reduced to a single integral by Lamarche and Beauchamp [17] and by Bandos et al. [18]. An extension of the solution to the case in which the BHE top is buried under the ground surface has been worked out by Claesson and Javed [19]. By finite-element simulations, Zanchini and Lazzari [20] determined dimensionless polynomial expressions that yield the values of $\theta(r, \tau)$ for the finite cylindrical-source scheme.

Both the finite line-source scheme and the finite cylindrical-source scheme with uniform heat flux per unit length yield some inaccuracies in the evaluation of $\theta_f(\tau)$, both for very short values of time (some minutes or few hours), and for very high values of time (many years).

The inaccuracy for low values of time is due to the fact that the FCS scheme completely neglects the thermal inertia of the BHE, and the FLS scheme approximates it by that of a cylinder with the same thermal properties as the ground. The inaccuracy for high values of time is due to the fact that, in real applications, the BHEs are fed in parallel with the same inlet temperature and the heat flux per unit length exchanged with the ground is not uniform. Dimensionless thermal response factors with the boundary condition first employed by Eskilson [21], namely time-constant total heat flux and uniform surface temperature of the BHE field, were determined analytically by Cimmino and Bernier [22] and numerically by Monzó et al. [23] and by Naldi and Zanchini [24]. In [22], the BHEs are divided into segments releasing different time-dependent heat fluxes per unit length, such that the surface temperature of the BHE field is uniform at each instant of time, and the analytical FLS solution is adopted for each segment. In [23], the BHEs are supposed to be filled with a High-Conductive Material (HCM), which also interconnects the BHEs and ensures a uniform temperature of the external surface of the BHE field. In [24], the condition of uniform surface temperature and time-constant total heat flux is obtained by iterations, through the convergence to time-constant total heat flux of the conditions of uniform and time-dependent surface temperatures.

The results obtained in [22–24] show that the assumption of uniform heat flux yields an overestimation of the thermal response factor with respect to the assumption of uniform surface temperature. The latter, however, yields an underestimation of the thermal response factor with respect to the real condition at the BHE boundary, that is intermediate between uniform heat flux and uniform temperature. Thermal response factors that are accurate for high values of time have been obtained analytically by Cimmino [25] and numerically by Puttige et al. [26] and by Monzó et al. [27], by taking into account the non-uniformity of the BHE surface temperature caused by the BHE thermal resistance. In [26,27] the result is obtained by introducing a thin resistive layer between the HCM and the ground.

The models described above do not predict accurately the BHE thermal response during the first working hours, because they do not consider the effects of the real internal structure of each BHE. Therefore, several authors developed BHE models suitable to determine accurately the short-term thermal response. In some models, the BHE is sketched as a thermal network, that includes both thermal resistances and heat capacities.

De Carli et al. [28] developed a model, called Capacity Resistance Model (CaRM), that employs lumped capacities and thermal resistances, and can be used for single U-tube, double U-tube and coaxial BHEs. The model was improved by Zarrella et al. [29] by taking into accounts also the heat capacity of the sealing grout and that of the BHE fluid. Other thermal resistance and capacity models (TRCMs) that consider the heat capacity of the grout were developed by Bauer et al. [30] for coaxial, single U-tube, and double U-tube BHEs. Pasquier and Marcotte [31] improved the TRCM developed in Ref. [30] by including the thermal capacities of the fluid and of the pipes, however the improved model neglects the finite length of the borehole. Ruiz-Calvo et al. [32] developed a thermal network model where the BHE is discretized vertically, and five thermal capacities and six thermal resistances are taken into account at each depth, considering the thermal properties of the ground, the grout and the pipes. The model was implemented in TRNSYS and was validated by comparison with the measurement outcomes of two distributed thermal response tests performed in Sweden.

Other authors analyzed the short-term BHE thermal response by employing the infinite line-source model in composite media. Li and Lai [33] developed an analytical solution that considers the different properties of the BHE material and of the ground, but disregards the finite length of the BHE, so that it cannot be applied to determine the long-term response. The same authors [34] recommended to divide into three parts the integrals that appear in the solution in order to perform the numerical integration. Li et al. [35] presented a full-time-scale temperature response function obtained by combining the short-term solution given in [33], the infinite line-source solution and the finite line-source solution. Zhang et al. [36] developed a quasi 3D line-source model to determine the full-time-scale evolution of the mean fluid temperature, that introduces a time-dependent BHE thermal resistance.

Several authors developed analytical or numerical cylindrical models to determine the short-term thermal response of U-tube BHEs. Xu and Spitler [37] proposed a numerical cylindrical model composed of an inner fluid annulus, an equivalent convective resistance layer, a tube layer, a grout layer, and the surrounding ground. The heat capacity and the thermal resistance of each layer are equal to those of the corresponding element of the real BHE. Lamarche and Beauchamp [38] developed an analytical model where the BHE is replaced by a cylindrical layer with the same thermophysical properties as the grout, with external radius $r_b$ and internal radius such that the thermal resistance of the grout layer is equal to the BHE thermal resistance. The model does not consider the heat capacity of the fluid. Lamarche [39] presented an analytical model improved with respect to that of [38], that reproduces the heat capacity and the thermal resistance of each BHE element. In the new model, the borehole is composed of a solid cylinder representing the fluid, where a given generation power per unit length is supplied, surrounded by a thin cylindrical layer representing the polyethylene pipes, surrounded on turn by a cylindrical layer representing the grout. The analytical solutions in [38,39] were determined by the Laplace transform method.

Bandyopadhyay et al. [40,41] determined the analytical solution, in the Laplace transformed domain, for a BHE model where the fluid is represented by a virtual solid cylinder, with the same heat capacity as the real fluid and a very high thermal conductivity, that generates heat uniformly. The fluid cylinder is surrounded by a grout layer, and by the ground. The solution in the Laplace transformed domain is inverted through the Gaver-Stehfest numerical algorithm.

Man et al. [42] developed two analytical solutions for a cylindrical BHE model where the BHE is sketched as a solid cylinder with the same thermal properties as the ground, containing a cylindrical heat source with negligible thickness and heat capacity that generates a uniform and constant thermal power. The first solution is 1D, while the second is 2D axisymmetric and considers the finite length of the BHE. The analytical solutions are validated by comparison with the line-source and the hollow cylindrical-source models and with the results of numerical computations.

Javed and Claesson [43] presented an analytical solution for a BHE model similar to that considered in [40,41]. The fluid is represented by a solid cylinder having a uniform temperature and the heat capacity of the BHE fluid, where a constant thermal power is injected. The solid cylinder is surrounded by a grout layer having the thermal properties of the grout, that is surrounded by the ground. A thermal

resistance equal to the pipe thermal resistance, including the convective resistance, is inserted between fluid and grout. The heat transfer problem is represented as a thermal network in the Laplace domain and the solution is given in the time domain in the form of an integral. The analytical solution is validated by comparison with a numerical one and with experimental results. Claesson and Javed [19] presented a method to determine the full-time-scale thermal response of a BHE field. Up to a certain time, the short-term thermal response obtained by the method of [43] is used. Later on, a generalized finite line-source solution for the ground is used, that considers also the BHE buried depth and is reduced to a single integral. The long-term response of the ground is shifted upwards to match the fluid temperature given by the short-term response at the breaking time.

Due to the difficulty of implementing a model that yields both the short-term and the long-term thermal response of a BHE field, some authors recommend using separate models [19,32], matched at a selected time instant, and employing the FLS model with uniform thermal power per unit length for the long-time response [19,35]. However, the latter model is insufficient to yield accurately the long-term response of a BHE field under the conditions of BHEs fed in parallel with the same inlet temperature and time-constant total heat flux.

In this paper, we show that it is possible to determine accurately the full-time-scale thermal response of a BHE field by employing one model, suitable for the short-term analysis and simple enough to be employed for long-term simulations. The model selected is the numerical cylindrical BHE model developed in [44], where a BHE is sketched by a One-Material Equivalent Cylinder (OMEC), with radius $r_b$, that contains an internal heat-generating surface with radius $r_{eq}$. The real condition of BHEs fed in parallel with the same inlet temperature and time-constant total heat flux is reached by repeated simulations with a uniform and time-dependent temperature of the heat-generating surface with radius $r_{eq}$. The convergence to a time-constant total heat flux is obtained by the method proposed in [24]. We show that the OMEC model is both very accurate in the short term and easily employable in long-term numerical simulations of BHE fields.

## 2. OMEC of a U-tube BHE

In this section we briefly describe the OMEC model developed in [44], and we optimize the model to determine the short-term thermal response of a selected single U-tube BHE. Then, we compare the short-term fluid-to-ground thermal response yielded by the OMEC with that obtained by a simulation of the real BHE and those obtained by applying typical BHE models, namely the FCS scheme and the High-Conductive Material (HCM) scheme [23,27], both complemented with a thin resistive layer at the BHE boundary. Since in this section we are considering the short-term thermal response, we perform 2D numerical simulations of the cross section, both for the real BHE and for the models.

In the simulation of the real BHE, the borehole fluid (water) is modeled as a solid that generates heat uniformly, with a high thermal conductivity, i.e., 1000 W/(m K). A high value of the thermal conductivity is used to achieve a nearly uniform temperature distribution in the fluid region. Moreover, the thermal conductivity of the polyethylene pipes is replaced by an effective one to include the convective thermal resistance. The mean-fluid-temperature rise, $\theta_f$, is evaluated as surface average in the fluid domain.

In the OMEC simulation, the borehole is modeled as a cylinder, with radius equal to the BHE radius, $r_b$, that contains a heat-generating cylindrical surface with an equivalent radius $r_{eq}$. The OMEC is made of an equivalent material, with the same heat capacity as the borehole, and with thermal conductivity such that the thermal resistance of the cylindrical layer between $r_{eq}$ and $r_b$ is the same as that of the BHE, $R_b$. The value of $R_b$ is evaluated through a numerical steady-state simulation of the borehole cross section including a portion of the surrounding ground, as recommended by Lamarche et al. [45] and by Zanchini and Jahanbin [46]. The convection coefficient is evaluated through the Churchill correlation with uniform wall heat flux [47]. The time-dependent value of $\theta_f$ is obtained as line average on the circumference with radius $r_{eq}$. In order to determine the optimal value of $r_{eq}$, finite-element simulations of transient heat conduction are performed for the OMEC by employing

trial values of $r_{eq}$, accurate up to 0.1 mm. The mean-fluid-temperature rise is compared with that evaluated by the simulation of the real BHE cross section. The value of $r_{eq}$ is then modified and the OMEC simulation is repeated until the Weighted Root Mean Square Deviation, *WRMSD*, during the first hour (the most critical one) is minimized. The *WRMSD* is defined as:

$$WRMSD = \sqrt{\frac{\sum_i \Delta\tau_i \left(\theta_{fO} - \theta_{fr}\right)_i^2}{\sum_i \Delta\tau_i}}, \tag{3}$$

where $(\theta_{fO} - \theta_{fr})_i$ is the difference between the mean-fluid-temperature rise obtained by the OMEC and that obtained by the real BHE, at the *i*-th time instant $\tau_i$, and $\Delta\tau_i$ is $(\tau_{i+1} - \tau_{i-1})/2$.

The accuracy of the OMEC is compared with that of the FCS model complemented with a thin resistive layer of thickness $t$ = 5 mm, placed on the BHE surface. In the FCS scheme, the BHE is modeled as a hollow cylinder with radius $r_b$, that delivers a uniform and constant thermal power per unit length, $q_l$, to the surrounding ground. The resistive layer has a thermal conductivity such that its thermal resistance is equal to that of the BHE and has a vanishing heat capacity. The power per unit length $q_l$ is applied on the inner surface of this layer, namely on the circumference of radius $r_l = r_b - t$, and the mean-fluid-temperature rise, $\theta_f$, is evaluated as line average on the circumference of radius $r_l$.

The accuracy of the OMEC is compared also with that of the HCM model employed by Monzó et al. [27]. In this model, the borehole is composed of a solid cylinder with a very high thermal conductivity, surrounded by a thin resistive layer. In our simulation, the high-conductive cylinder has radius $r_l$ and thermal conductivity $10^6$ W/(m K), while the thin resistive layer has internal radius $r_l$, external radius $r_b$, thickness $t$ = 5 mm and thermal conductivity such that the thermal resistance of the layer is equal to that of the BHE. The high-conductive cylinder and the thin resistive layer have the same volumetric heat capacity, such that the sum of their heat capacities equals that of the borehole. The time-dependent value of $\theta_f$ is evaluated as surface average on the circle of radius $r_l$.

All these models reproduce the borehole thermal resistance, thus they are very accurate in the long term, where the BHE thermal resistance is the only relevant parameter. Thus, in order to compare the accuracy of the models, we perform short-term simulations of a BHE cross section, surrounded by a ground layer with external radius $r_g$ = 5 m, for a time-period of 100 h. We select the single U-tube BHE illustrated in Figure 1, with radius $r_b$ = 75 mm, shank spacing $s$ = 94 mm, and polyethylene tubes with external radius $r_{po}$ = 20 mm and internal radius $r_{pi}$ = 16.3 mm. The BHE is subjected to a uniform and constant thermal power per unit length $q_l$ = equal to 50 W/m. The thermal conductivity of the grout is $k_{gt}$ = 1.6 W/(m K) and that of the ground is $k_g$ = 1.8 W/(m K). The volumetric heat capacity of the grout is $(\rho c)_{gt}$ = 2.5 MJ/(m$^3$ K) and that of the ground is $(\rho c)_g$ = 3.0 MJ/(m$^3$ K). The BHE fluid, that is water, has a volume flow rate of 14 L/min. The water properties are evaluated at 20 °C, through the NIST website [48]. The convection coefficient, obtained by the Churchill correlation at constant wall heat flux [47], is 1472 W/(m$^2$ K). The BHE thermal resistance, evaluated by a steady-state 2D finite-element simulation of the real BHE cross section including a ground layer with $r_g$ = 5 m, is $R_b$ = 0.09788 m K/W.

The equation of transient heat conduction to be solved in the *i*-th material is

$$(\rho c)_i \frac{\partial T_i(r, \tau)}{\partial \tau} = k_i \nabla^2 T_i(r, \tau) + q_{geni}, \tag{4}$$

where the subscript *i* denotes the *i*-th material, *r* is the radial coordinate, $\tau$ is time, and $q_{gen}$ is the power generated per unit volume. The initial condition is:

$$T_i(r, 0) = T_g. \tag{5}$$

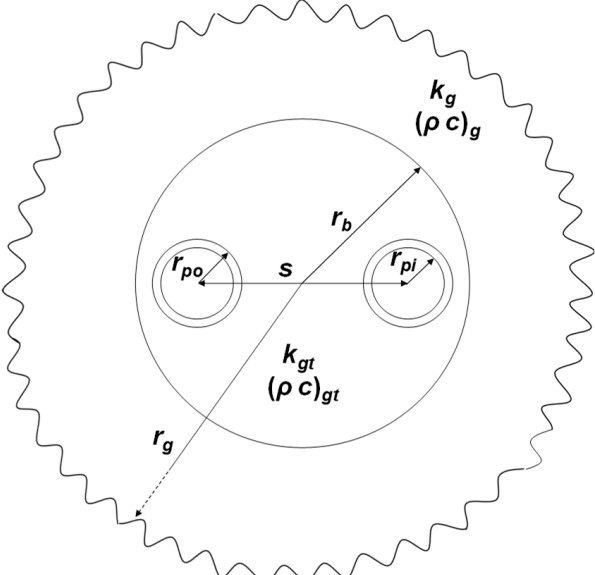

**Figure 1.** Sketch of the selected BHE cross section, with surrounding ground.

At the outer boundary of the ground, an adiabatic condition is imposed:

$$\left. \frac{\partial T_i(r, \tau)}{\partial r} \right|_{r = r_g} = 0. \tag{6}$$

In the simulation of the real BHE cross section, a uniform generation term is imposed in the fluid domain:

$$q_{geni} = \frac{q_l}{2 \pi r_{pi}^2}; \tag{7}$$

in the simulation of the OMEC cross section, a uniform surface heat source is imposed on the circumference of radius $r_{eq}$:

$$q_s = \frac{q_l}{2 \pi r_{eq}}; \tag{8}$$

in the simulation of the FCS cross section, a uniform heat flux is imposed at the inner surface of the thin resistive layer:

$$q = \frac{q_l}{2 \pi r_l}; \tag{9}$$

in the simulation of the HCM cross section, a uniform generation term is imposed in the high-conductive cylinder:

$$q_{geni} = \frac{q_l}{\pi r_l^2}. \tag{10}$$

A transient 2D finite-element simulation is performed, for the real BHE and for the three models, by using the software COMSOL Multiphysics. The working time is 100 h, with steps of 0.05 in the logarithm of time in seconds. A structured grid is built for each simulation, with about 5800 elements for the real BHE, about 2800 elements for the OMEC, about 1300 elements for the FCS and about 2800 elements for the HCM. Obviously, the higher number of mesh elements for the real BHE simulation is due to the complex geometry of the BHE internal structure, and the lower number of mesh elements for the FCS simulation is because the cylinder is hollow. Particulars of the grids adopted for the simulations are illustrated in Figure 2. The relative tolerance and the absolute tolerance are set in COMSOL equal to 0.0001. The computation time is about 10 s for the simulation of the real BHE and about 5 s for the simulations of the cylindrical models, on a PC with Intel Core i7-6700K 4.0 GHz and

RAM 64 GB. The optimized value of the OMEC $r_{eq}$, obtained by minimizing the *WRMSD* from the values of $\theta_f$ given by the simulation of the real BHE, is 22.0 mm.

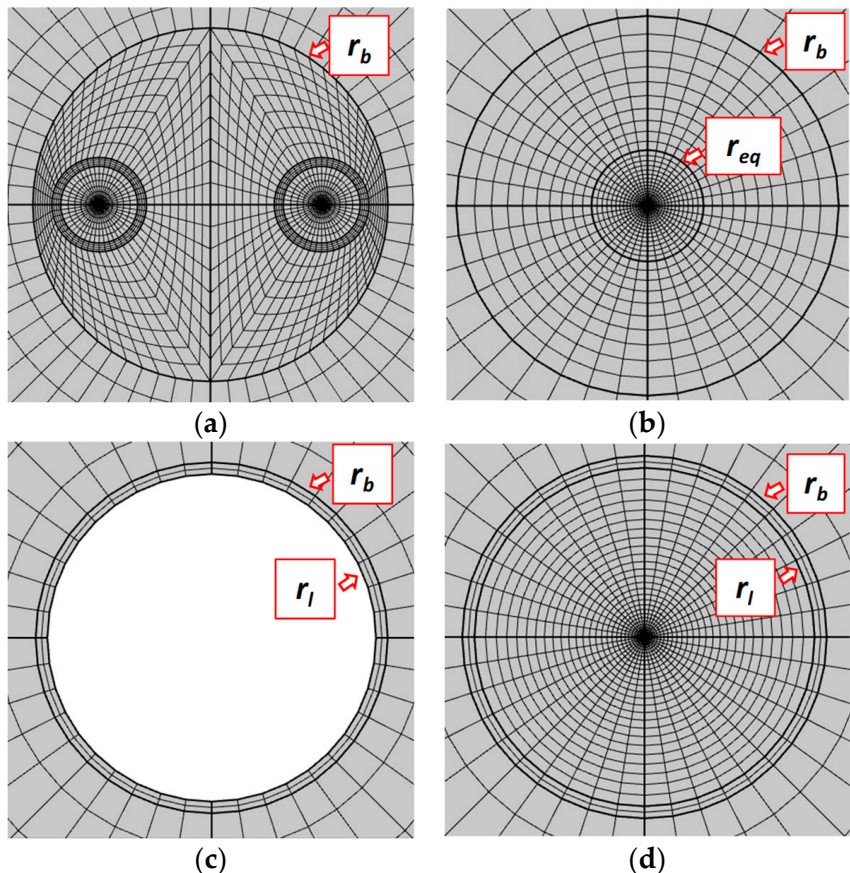

**Figure 2.** Particular of the mesh for the simulation of: (**a**) the real BHE cross section; (**b**) the OMEC cross section; (**c**) the FCS cross section; (**d**) the HCM cross section.

We checked that the results are independent of the domain extension by replacing the adiabatic condition (6) by a condition of uniform and constant temperature equal to $T_g$. The mesh independence of results was checked by replacing the grid with one regularly refined, with 4 times more elements. For both checks, nearly identical results were obtained, so that the independence of the domain extension and of the grid is ensured.

The mean-fluid-temperature rise obtained by each simulation is plotted versus the logarithm of time in seconds in Figure 3. The figure highlights that the OMEC is the most accurate model, being able to reproduce exactly the temperature rise given by the real BHE simulation for most of the time, with only a small discrepancy during the first 10 min. On the contrary, the FCS scheme shows the lowest accuracy. Indeed, this model does not take into account the BHE thermal capacity, that dampens the temperature rise of the BHE fluid, so that the model overestimates the values of $\theta_f$. The HCM model performs better than the FCS model, but worse than the OMEC, and underestimates the mean-fluid-temperature rise. Thus, both the FCS and the HCM model are not sufficiently accurate in the prediction of the short-term mean-fluid-temperature rise.

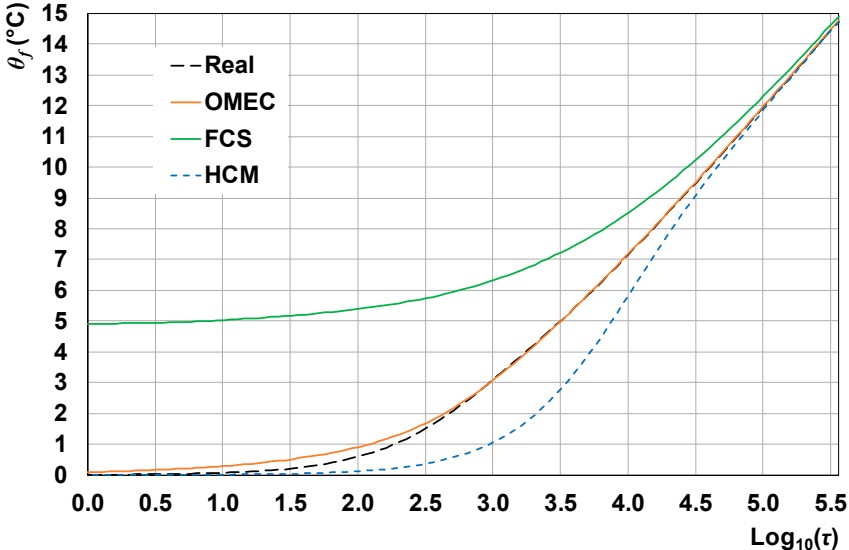

**Figure 3.** Mean-fluid-temperature rise versus the logarithm of time in seconds, for the real BHE and for the three BHE models.

## 3. Thermal Response of a 2 × 2 Borefield with Uniform Fluid Temperature

In this section we employ the OMEC model to evaluate numerically the full-time-scale fluid-to-ground thermal response of a 2 × 2 borehole field, with isothermal fluid. The values of the BHE geometrical and physical parameters reported in Section 2 are used. In addition, each BHE of the field has length $H = 100$ m, is buried at a depth $B_d = 2$ m from the ground surface and is placed at a distance $d = 6$ m from the adjacent BHEs. The physical properties of the ground are the same as in Section 2. A horizontal and a vertical view of the 2 × 2 square field are illustrated in Figure 4.

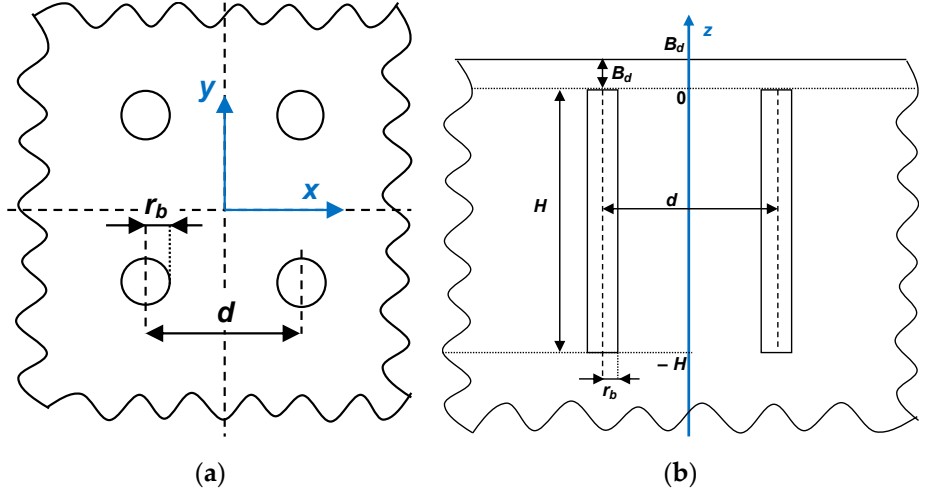

(a)                                      (b)

**Figure 4.** (a) Horizontal view of the 2 × 2 borefield; (b) Vertical view of the 2 × 2 borefield.

Only a quarter of the borefield is simulated, thanks to the field double symmetry. A cube with side 300 m is chosen as 3D computational domain. The domain includes one borehole, modeled as an equivalent cylinder through the OMEC method. In order to compact the domain, all the vertical lengths are reduced by a factor 5, and the thermal conductivities of the OMEC and of the ground in the vertical direction are reduced by a factor 25, according to the method employed by Zanchini et al. [49]. The geometry of the domain is shown in Figure 5.

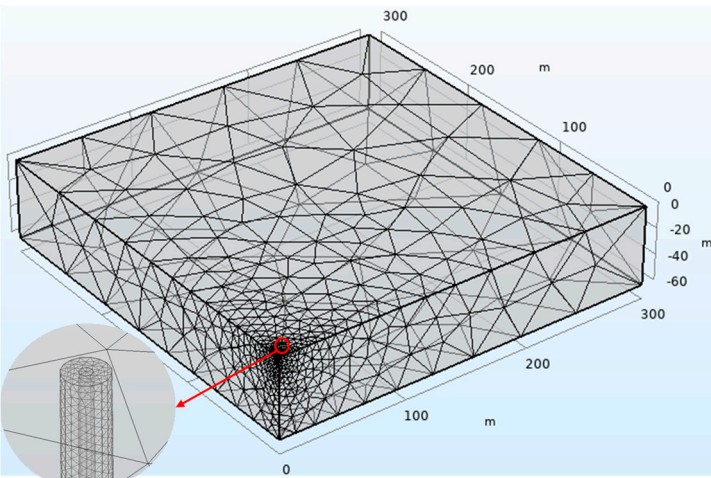

**Figure 5.** Computational domain, with mesh, for the 2 × 2 borefield.

The outer vertical surfaces and the bottom horizontal surface of the domain are set as adiabatic, a symmetry boundary condition is employed at the vertical symmetry planes, and the upper horizontal surface is set as isothermal, with $T = T_g$. On the OMEC heat-generating cylindrical surface of radius $r_{eq}$, an isothermal condition is imposed, with time-dependent temperature values evaluated through recursive simulations, as explained later. The initial condition is that of uniform temperature equal to $T_g$. The selected mesh is unstructured with about 680,000 tetrahedral elements. Finer elements are employed near the OMEC, while coarser ones are used towards the external boundaries of the domain. Figure 5 shows the mesh employed, including a zoom of the BHE top. Simulations are performed for a working time of $10^{10}$ s (about 317 years), with time steps in the logarithm of time in seconds equal to 0.05. The relative and absolute tolerances are 0.0001.

The recursive procedure developed by the authors [24] to evaluate the isothermal *g-function* of a borefield is here modified in order to calculate the fluid-to-ground thermal response of the 2 × 2 field, with isothermal fluid and with a time-constant total thermal power supplied to the fluid. The first simulation of the recursive procedure employs a uniform and time-constant heat source per unit area, $q_s$, at the cylindrical surface of radius $r_{eq}$, with value $q_s = 50/(2\pi r_{eq})$ W/m$^2$. The mean-fluid-temperature rise $\theta_f$ obtained, that will be called isoflux fluid-to-ground thermal response, is plotted in Figure 6 (Mesh 1) as a function of the logarithm of time in seconds.

The domain-size independence of the results is checked by repeating the simulation with the isothermal condition $T = T_g$ at the external boundaries of the computational domain. Nearly identical results are obtained. The mesh independence of the results is checked by repeating the simulation with a finer unstructured mesh, having about 1,800,000 elements. The time evolution of $\theta_f$ calculated by the finer mesh is illustrated in Figure 6 (Mesh 2). The results obtained by the two meshes are very similar, with a small discrepancy at the last time instants, evidenced by the zoom of Figure 6. The highest difference between the values of $\theta_f$ determined by the coarser mesh and those obtained by the finer mesh is +0.09 °C. Since the results are very similar, and the simulation time is about 2 h with the coarser mesh and about 12.5 h with the finer mesh, on a PC with Intel Core i7-6700K 4.0 GHz and RAM 64 GB, the coarser grid is employed.

The isoflux fluid-to-ground thermal response yielded by the first simulation of the recursive procedure, that will be denoted by $\theta_{f1}$, is used in the second simulation as a time-dependent isothermal condition on the OMEC cylindrical surface with radius $r_{eq}$. Then, one calculates the thermal power per unit length released by the cylindrical surface with radius $r_{eq}$ under this boundary condition, averaged along the borefield and averaged from the initial time instant to the current time instant. One evaluates the time-dependent ratio between this value and the desired time-constant value, namely $\bar{q}_l = 50$ W/m. The inverse of this ratio is used as a time-dependent multiplicative correction factor for

$\theta_{f1}$, to obtain the first-trial fluid-to-ground thermal response of the field with isothermal fluid, denoted by $\theta_{f2}$. The values of $\theta_{f2}$ are then employed in the third simulation as a time-dependent isothermal condition at the cylindrical surface with radius $r_{eq}$. The new time-dependent multiplicative correction factor for $\theta_{f2}$ is determined, to evaluate the second-trial isothermal fluid-to-ground thermal response, $\theta_{f3}$, and so on. The iterations can be stopped when the correction factor becomes sufficiently close to 1. In the present work, three isothermal fluid-to-ground thermal responses, $\theta_{f2}$, $\theta_{f3}$, $\theta_{f4}$, are evaluated.

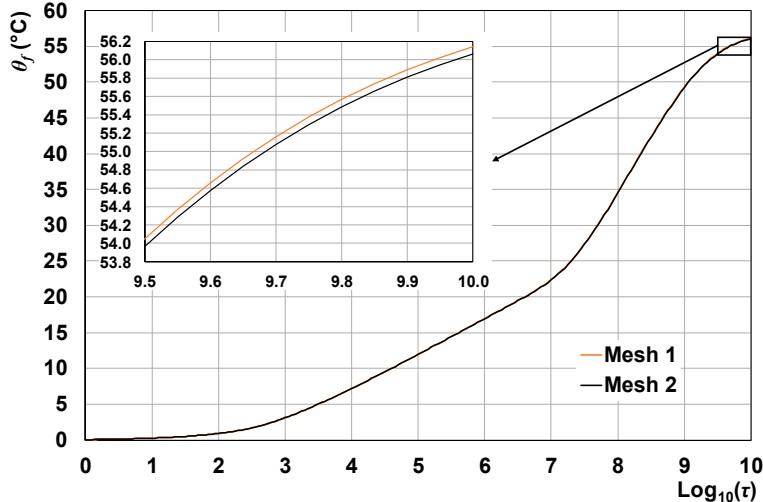

**Figure 6.** Isoflux fluid-to-ground thermal response of the $2 \times 2$ borefield versus $\log_{10}(\tau)$, with Mesh 1 (680,000 elements) and with Mesh 2 (1,800,000 elements).

The fluid-to-ground thermal response of the field determined with the isoflux condition, $\theta_{f1}$, and those determined with the isothermal conditions, $\theta_{f2}$, $\theta_{f3}$, $\theta_{f4}$, are plotted versus the logarithm of time in seconds in Figure 7. All the curves are practically coincident in the short term. On the contrary, employing the condition of uniform and time-constant heat source at the cylindrical surface of radius $r_{eq}$ yields an overestimation of the fluid-to-ground thermal response in the long term (compare the orange and the red line in Figure 7). Indeed, at the final time instant, $\theta_{f1}$ is about 2 °C higher than $\theta_{f4}$, as highlighted by the zoom of Figure 7. Moreover, the zoom evidences the negligible difference between $\theta_{f3}$ and $\theta_{f4}$, so that further iterations would be useless.

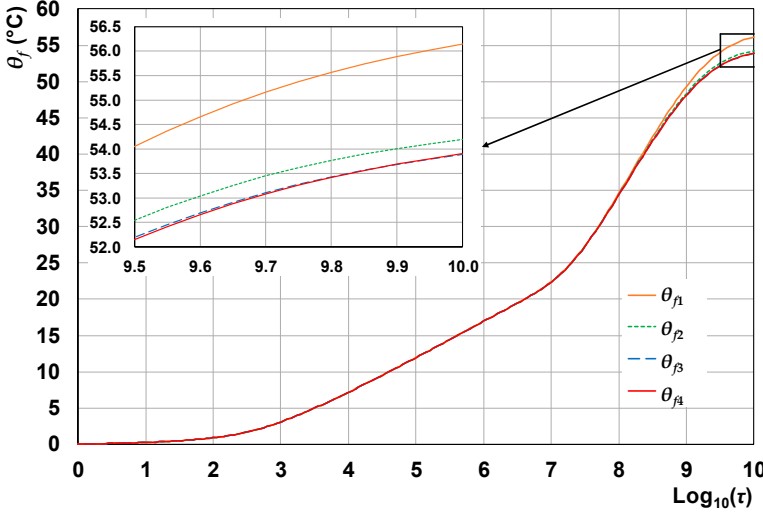

**Figure 7.** Fluid-to-ground thermal responses of the $2 \times 2$ borefield versus $\log_{10}(\tau)$, with the isoflux condition ($\theta_{f1}$) and with the isothermal conditions ($\theta_{f2}$, $\theta_{f3}$, $\theta_{f4}$), OMEC model.

The time evolutions of the length-averaged thermal power per unit length, $\bar{q}_{l1}$, $\bar{q}_{l2}$, $\bar{q}_{l3}$, $\bar{q}_{l4}$, yielded respectively by the thermal responses $\theta_{f1}$, $\theta_{f2}$, $\theta_{f3}$, $\theta_{f4}$, are plotted in Figure 8. The figure shows the convergence towards the desired time-constant value, $\bar{q}_l = 50$ W/m. The last isothermal thermal response, $\theta_{f4}$, yields a length-averaged thermal power per unit length very close to the desired value, with a maximum discrepancy of +0.05 W/m in the time interval $0 \le \log_{10}(\tau) \le 10$.

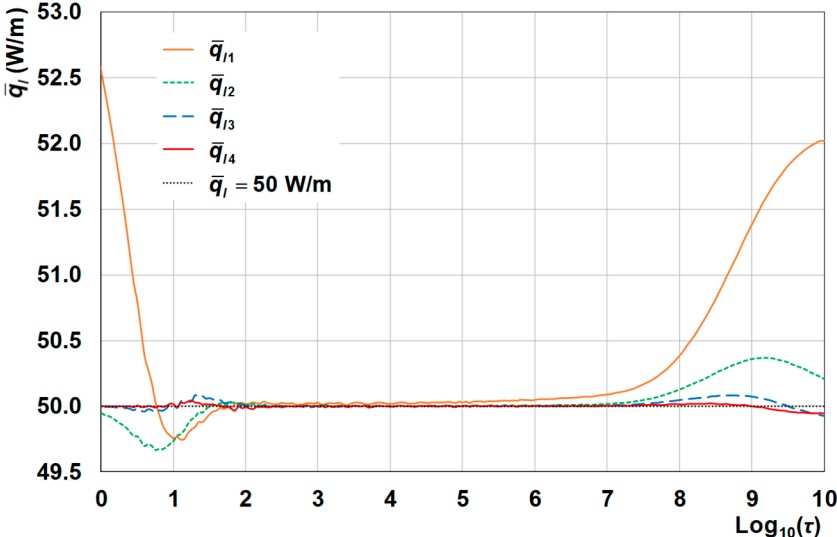

**Figure 8.** Plots of $\bar{q}_l$ versus $\log_{10}(\tau)$, for the fluid-to-ground thermal responses $\theta_{f1}$, $\theta_{f2}$, $\theta_{f3}$, $\theta_{f4}$, OMEC model.

The fluid-to-ground thermal response of the $2 \times 2$ borefield with isothermal fluid obtained through the OMEC model is compared with that obtained through the FCS model with thin resistive layer, that is the simplest model yielding an accurate thermal response in the long term. The recursive procedure to determine the isothermal response of the field is now performed by imposing the isothermal conditions at the inner surface of the thin resistive layer, where the mean-fluid-temperature rise is evaluated. For each simulation, the computation time is the same as that for the OMEC. The fluid-to-ground thermal responses of the borefield, $\theta_{f1}$, $\theta_{f2}$, $\theta_{f3}$, $\theta_{f4}$, are plotted versus $\log_{10}(\tau)$ in Figure 9.

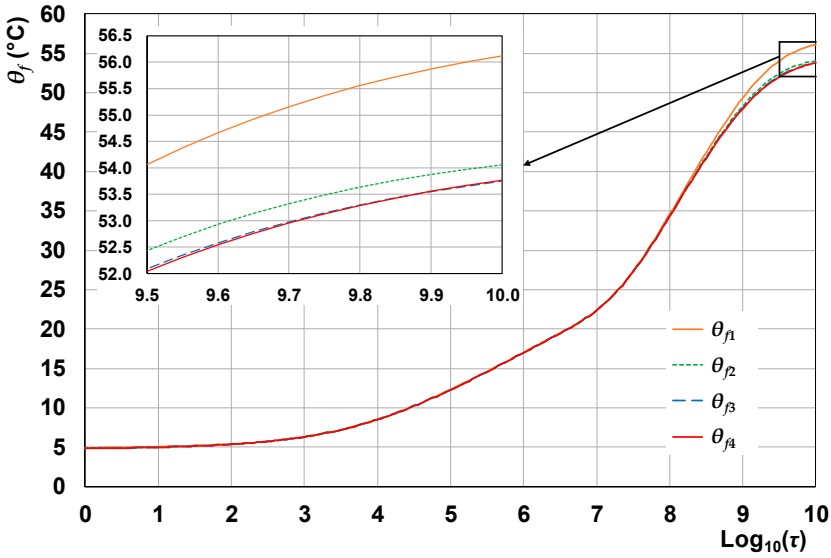

**Figure 9.** Fluid-to-ground thermal responses of the $2 \times 2$ borefield versus $\log_{10}(\tau)$, with the isoflux condition ($\theta_{f1}$) and with the isothermal conditions ($\theta_{f2}$, $\theta_{f3}$, $\theta_{f4}$), FCS model.

As for the OMEC, the isoflux condition yields an overestimation of the fluid-to-ground thermal response in the long term, and the difference between $\theta_{f3}$ and $\theta_{f4}$ is negligible. Figure 10 shows the convergence of the length-averaged linear heat flux at the inner surface of the thin resistive layer towards the desired time-constant value $\bar{q}_l = 50 \text{ W/m}$.

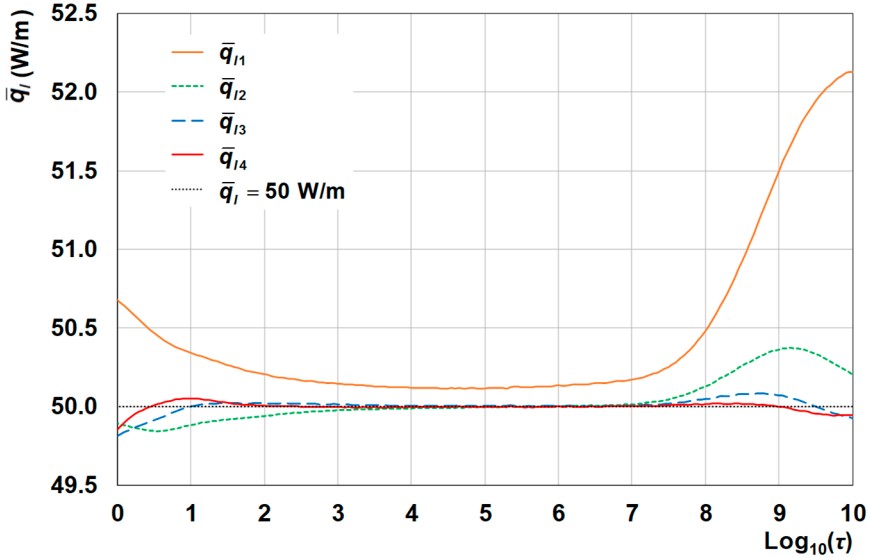

**Figure 10.** Plots of $\bar{q}_l$ versus $\log_{10}(\tau)$, for the fluid-to-ground thermal responses $\theta_{f1}$, $\theta_{f2}$, $\theta_{f3}$, $\theta_{f4}$, FCS model.

The fluid-to-ground thermal response of the borefield with isothermal fluid obtained through the OMEC model is compared with that obtained through the FCS model in Figure 11. The two curves are almost coincident in the long term, where the BHE thermal resistance only has an effect. The discrepancy at the last time instant, evidenced by the zoom of Figure 11, is only 0.14 °C. In the short term, where the BHE internal structure is important, the FCS scheme strongly overestimates the fluid temperature rise, up to 4.79 °C.

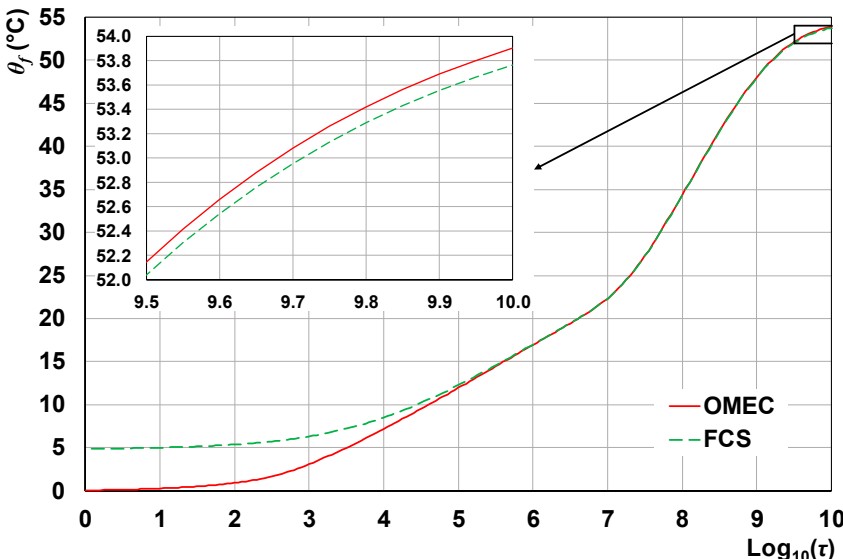

**Figure 11.** Isothermal fluid-to-ground thermal response of the $2 \times 2$ borefield versus $\log_{10}(\tau)$, OMEC model and FCS model.

## 4. Validation of the Simulation Code

The accuracy of the 3D finite-element code adopted for the OMEC simulation is checked by comparison with the analytical solution by Man et al. [42], who modeled the borehole as a finite solid cylinder, with the same thermal properties as the ground, that contains a heat-generating cylindrical surface with radius $r_{eq}$. The expression of the length-averaged temperature rise, $\theta$, is:

$$
\begin{aligned}
\theta(r,\tau) \;=\; & \frac{q_l}{8H\pi k_g}\int_0^H\int_0^\tau \frac{1}{\tau-\varphi}I_0\!\left[\frac{r\,r_{eq}}{2(\tau-\varphi)k_g/(\rho c)_g}\right] \\
& \exp\!\left[\frac{r^2+r_{eq}^2}{4(\tau-\varphi)k_g/(\rho c)_g}\right]\!\left\{erf\!\left[\frac{H-z}{2\sqrt{(\tau-\varphi)k_g/(\rho c)_g}}\right]\right. \\
& +2erf\!\left[\frac{z}{2\sqrt{(\tau-\varphi)k_g/(\rho c)_g}}\right]-erf\!\left[\frac{H+z}{2\sqrt{(\tau-\varphi)k_g/(\rho c)_g}}\right]\!\left.\right\}d\varphi dz
\end{aligned}
\tag{11}
$$

where $I_0$ is the modified Bessel function of first kind and zero order, *erf* is the error function and $z$ is the vertical coordinate.

We consider a 2 × 2 borefield with the same values of $r_b$, $H$, $B_d$, and $d$ as that simulated in Section 3, and sketch each BHE by a OMEC with $k_O = k_g = 1.8$ W/(m K), $(\rho c)_O = (\rho c)_g = 3.0$ MJ/(m³ K), and $r_{eq} = 22.0$ mm. We perform a 3D simulation of the field in COMSOL by imposing a uniform and time-constant heat source per unit area, $q_s = 50/(2\pi r_{eq})$ W/m², on the OMEC heat-generating cylindrical surface. The time evolution of the mean-fluid-temperature rise is shown in Figure 12, red line.

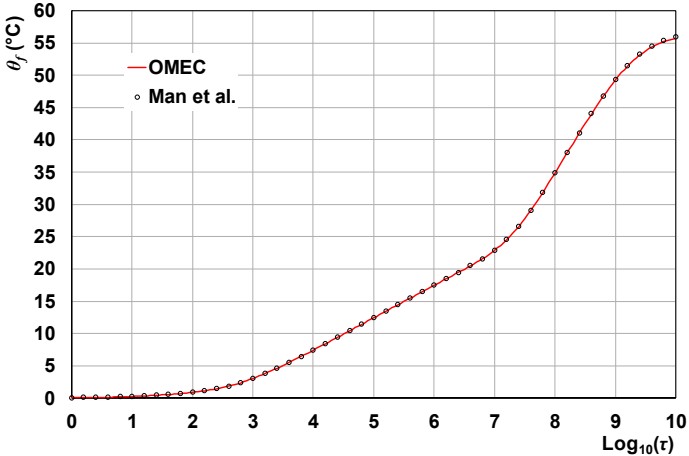

**Figure 12.** Isoflux fluid-to-ground thermal response of a 2 × 2 borefield versus $\log_{10}(\tau)$, simulation of OMEC and analytical solution by Man et al. [42].

Then, we employ Equation (11) to evaluate the time-dependent values of $\theta$ at the following radial distances from the BHE axis: $r = r_{eq} = 22.0$ mm, $r = d = 6$ m, $r = d\sqrt{2} = 6\sqrt{2}$ m. Indeed, the isoflux fluid-to-ground thermal response of the field can be obtained by employing the superposition of the effects of the single BHEs: the mean-fluid-temperature rise in a BHE of the field can be evaluated as the sum of the temperature rises due to the BHE itself and to the other BHEs of the field, each at its radial distance from the BHE considered. In the case of our 2 × 2 square field, the distance between two BHEs is either $d$ or $d\sqrt{2}$, so that the mean-temperature-rise of the field is given by:

$$
\theta_f(\tau) \;=\; \theta(r_{eq},\tau)+2\,\theta(d,\tau)+\theta(d\sqrt{2},\tau).
\tag{12}
$$

The values of $\theta_f$ obtained through the analytical solution by Man et al., at several time instants in the range $0 \le \log_{10}(\tau) \le 10$, are reported in Figure 12, by black circles.

Figure 12 shows a good agreement between the values of $\theta_f$ calculated through the numerical simulation of the OMEC and those obtained through the analytical expression by Man et al. [42].

The root mean square deviation is 0.062 °C in the whole time range, whereas it is 0.023 °C in the range $0 \leq \log_{10}(\tau) \leq 8$. The maximum relative difference between the numerical values and those obtained through Equation (11) is −0.40%, and occurs at the last time instant. Due to the very low differences, the numerical code employed in our simulations can be considered as validated.

## 5. Validation of the Model by Comparison with Experimental Results

In order to validate the OMEC model by comparison with experimental data, we consider the results of a Thermal Response Test (TRT) with duration 110 h, performed on a double U-tube BHE located in Fiesso D'Artico (Venice, Italy), and evaluated numerically by accurate finite-element simulations of the real BHE cross section [50]. The BHE has length 100 m, radius 78 mm, shank spacing 83.45 mm, polyethylene tubes with external radius 16 mm and internal radius 13 mm. A mean thermal power per unit length of 72.67 W/m was delivered to the BHE through an external water tank with volume 98 L, and the water volume flow rate was 26.51 L/min. The values of the thermal properties of the grout and of the ground, estimated in Ref. [50] through the simulation of the real BHE cross section, are: $k_{gt}$ = 1.13 W/(m K), $(\rho\, c)_{gt}$ = 1.8 MJ/(m$^3$ K), $k_g$ = 1.77 W/(m K), $(\rho\, c)_g$ = 2.5 MJ/(m$^3$ K). The convection coefficient is 1864 W/(m$^2$ K) and the BHE thermal resistance is 0.0921 m K/W [50].

We perform a simulation of the real BHE cross section and repeated simulations of the OMEC model in order to optimize the value of $r_{eq}$; the result is $r_{eq}$ = 49.2 mm. The mesh employed for the simulation of the real BHE has about 14,000 elements, and that used for the simulations of the OMEC model about 2800 elements. Both the absolute and the relative tolerance are set equal to 0.0001, and the time step is 1 min. In each simulation, the computational domain has an external radius of 5 m and external adiabatic boundary. In the simulation of the real BHE, the initial condition is 14.5 °C for water and tubes (experimental value), 14.3 °C for the ground (measured value of $T_g$), and the intermediate value 14.4 °C for the grout. In the simulation of the OMEC, the initial condition is 14.5 °C for the cylinder and 14.3 °C for the ground. The effects of the thermal inertia of water, including that contained in the external tank, are evaluated as in [50].

The time evolutions of the mean fluid temperature, $T_f$, obtained by the simulations of the real BHE and of the OMEC are compared with the experimental values of $T_f$ in Figure 13. The figure shows an excellent agreement with the experimental data both for the real BHE simulation and for the OMEC simulation. The root mean square deviation from the measured values is 0.14 °C for the real BHE and 0.15 °C for the OMEC. Therefore, the OMEC models accurately the real BHE.

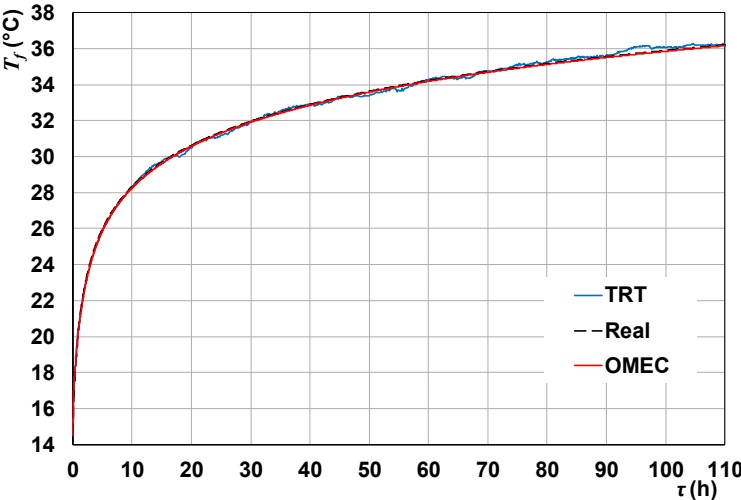

**Figure 13.** Plots of $T_f$ versus time yielded by the TRT [50] and by the simulations of the real BHE and of the OMEC.

## 6. Conclusions

We have presented a method to determine the full-time-scale fluid-to-ground thermal response of a borehole heat exchanger (BHE) field under the condition of uniform fluid temperature and time-constant total thermal power supplied to the fluid, that is necessary for the long-term accuracy. The method is simple and accurate, and employs one numerical model, instead of a short-term and a long-term model matched at a selected time instant. The method has been illustrated by determining, as an example, the fluid-to-ground thermal response of a 2 × 2 field of single U-tube BHEs.

Each BHE has been modeled as a one-material equivalent solid cylinder (OMEC), with the same length and radius as the BHE, surrounded by the ground and containing a heat-generating cylindrical surface whose radius is optimized by 2D numerical simulations of the BHE cross section. The temperature of the heat-generating surface represents the fluid temperature. The thermal conductivity and the volumetric heat capacity of the equivalent cylinder are such that the thermal resistance of the layer between the generating surface and the BHE external surface equals the BHE thermal resistance and the total heat capacity of the cylinder equals that of the BHE, including the fluid.

The condition of uniform fluid temperature and time-constant total heat flux has been reached by repeated 3D finite-element simulations of the BHE field, performed through the software COMSOL Multiphysics. Conditions of uniform and time-dependent temperature of the heat-generating surface are employed, that converge to a time-constant total heat flux.

Each 3D simulation by the OMEC required a computational time of about two hours on a PC equipped with an Intel Core i7-6700K 4.0 GHz CPU and 64 GB RAM. The same computation time was required for simulations with the finite cylindrical-source model with thin resistive layer, that yielded accurate results only in the long term.

The accuracy of the 3D simulation code by the OMEC has been proved by comparison with the results obtained by applying the analytical solution developed by Man et al. [42], in the special case of properties of the cylinder equal to those of the surrounding ground, and uniform heat source per unit area at the generating surface. The accuracy of the OMEC model has been proved by comparing the outcomes of the model with the experimental results of a thermal response test that was evaluated numerically by finite-element simulations of the real BHE cross section.

In future research work, the OMEC model and the recursive procedure to obtain the condition of uniform fluid temperature will be employed to provide accurate full-time-scale fluid-to-ground dimensionless thermal response factors of some kinds of BHE fields, such as in-line fields and square fields, with variable internal structure, radius and length of the BHEs, distance between the BHEs, number of BHEs in the field.

**Author Contributions:** Conceptualization, E.Z.; methodology, C.N. and E.Z.; software, C.N.; validation, C.N. and E.Z.; formal analysis, C.N.; investigation, C.N. and E.Z.; writing—original draft preparation, C.N. and E.Z.; writing—review and editing, C.N. and E.Z.; visualization, C.N.

**Funding:** This research received no external funding.

**Conflicts of Interest:** The authors declare no conflict of interest.

## Nomenclature

| | | |
|---|---|---|
| $B_d$ | Buried depth | (m) |
| BHE | Borehole Heat Exchanger | |
| $d$ | Borehole spacing | (m) |
| *erf* | Error function | |
| FCS | Finite Cylindrical-Source | |
| FLS | Finite Line-Source | |
| GCHP | Ground-Coupled Heat Pump | |
| HCM | High-Conductive Material | |
| *i* | Of the *i*-th material or *i*-th time instant | |
| $I_0$ | Modified Bessel function of first kind and zero order | |

| $H$ | Borehole length | (m) |
|---|---|---|
| $k$ | Thermal conductivity | (W/(m K)) |
| OMEC | One-Material Equivalent Cylinder | |
| $Q$ | Thermal power | (W) |
| $q$ | Heat flux per unit area | (W/m$^2$) |
| $q_{gen}$ | Generated thermal power per unit volume | (W/m$^3$) |
| $q_l$ | Thermal power per unit length | (W/m) |
| $q_s$ | Surface heat source | (W/m$^2$) |
| $r$ | Radius, radial coordinate | (mm) |
| $R_b$ | Borehole thermal resistance | (m K/W) |
| $s$ | Shank spacing | (mm) |
| $T$ | Temperature | (°C) |
| $t$ | Thickness of the thin resistive layer | (mm) |
| TRT | Thermal Response Test | |
| *WRMSD* | Weighted Root Mean Square Deviation | (°C) |
| $z$ | Vertical coordinate | |

**Greek symbols**

| $\theta$ | Temperature rise | (°C) |
|---|---|---|
| $\rho c$ | Volumetric heat capacity | (MJ/(m$^3$ K)) |
| $\tau$ | Time | (s) |

**Subscripts**

| $b$ | Borehole |
|---|---|
| $eq$ | Equivalent |
| $f$ | Fluid |
| $g$ | Ground |
| $gt$ | Grout |
| $l$ | Of the inner surface of thin layer |
| $O$ | Of the OMEC |
| $pi$ | Of the inner surface of pipe |
| $po$ | Of the outer surface of pipe |
| $r$ | Of the real BHE |

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
