# Peer review of "Full-Time-Scale Fluid-to-Ground Thermal Response of a Borefield with Uniform Fluid Temperature"

_energies, doi:10.3390/en12193750_

Round 1
Reviewer 1 Report
About the title: In my opinion, borefield or borehole field better than bore field, uniform fluid temperature better isothermal fluid (might seem synonymous but it's not the same.)
l49: BHE thermal resistance better than BHE thermal resistance per unit length.
l221-222: "in the long term, where only the effect of the BHE thermal resistance is important" I do not agree with this sentence. I think that the thermal resistance of the borehole has relevant importance during the transient time. Please, cite where that conclusion comes from.
eq 4: r is r_b
l255: the optimized value of r_eq: Where does it come from? Why that value? Explain in more detail or cire. Why is only that value optimized? Perhaps the differences in the models shown in figure 1 come because the values of the other models are not optimized for the real BHE with which it is compared. For example, is the resistive layer value of the FCS scheme optimized?
l289: if a quarter of the borefield is simulated according to Figure 3 (a), in Figure 4 the borehole has to be in (150,150) not in (0,0)
Differences between Figure 5 and Figure 2. Does OMC model have to be the same curve? I do not see where is included the thermal influence of the other bhe in the field 2x2
l371:validation: I think it would be very interesting to also validate the OMEC model with real Thermal Response Test and compare if the outputs of the model correspond to the real thermal behavior in a bhe. In our research group, we have many very precise thermal tests on single U bhe that can be very useful to validate this model.
l401 conclusions: the usefulness of the conclusions should be enhanced. If we obtain the same results with OMEC as with Man et.al. why use this numerical model which is more complex than an analytical model?
Figures 6, 7, 8, 9, there is no magnitude in the y-axis. In general, I don't like that magnitude is between values
PD: it is important to be able to review the article in the bibliography [39] as it is strongly linked to this article but is not accessible as it is still under review
Reviewer 2 Report
The manuscript is very interesting: the field, such as ground thermal response test is important. The numerical modelling of the full-time-scale thermal response test through COMSOL code is innovative and well detailed. The aim of the work is clearly stated and different layouts of BHEs are considered showing good results. The suggestion is to take in account the contribution of the groundwater flow: this contribution, depending on the flow velocity, could cause an important thermal impact in aquifer, changing the results of simulation. Therefore, the results achieved now in the paper could be representative of the only situation where the groundwater velocity is null. Moreover, a validation of experimental data related to numerical modelling or analytical solution could improve the quality of the paper, but this is only a suggestion for future scientific publications. The English language is good and clear. Here a list of detailed comments/suggestions:
Lines 57-59. The sentence needs a more detailed discussion.
Lines 68-72.
Lines 81-82. About papers discussing the grout sealing the borehole, I suggest to read and implement:
- Alberti L., Angelotti A., Antelmi M., La Licata I., 2017. A Numerical Study on the Impact of Grouting Material on Borehole Heat Exchangers Performance in Aquifers, Energies, 10, 703.
- Delaleux F., Py X., Olives R., Dominguez A., 2012. Enhancement of geothermal borehole heat exchangers performances by improvement of bentonite grouts conductivity. Applied Thermal Energy, 33–34, 92–99.
Introduction chapter: the chapter is too detailed and needs to be summarized (at instance, lines 109-135). I suggest to reduce the concepts about infinite line source and introduce some concepts about the actual solution of the moving line source, implementing the groundwater flow velocity. Please read and implement:
- Hecht-Mendez J, Molina-Giraldo N, Blum P, Bayer P. Evaluating MT3DMS for heat transport simulation of closed geothermal systems. Groundwater 2010; 5(48):741–56.
- Wang H, Qi C, Du H, Gu J 2009 Thermal performance of borehole heat exchanger under groundwater flow: a case study from Baoding Energy and Buildings 41 1368-1373
Please also consider the similar research activity where authors implement the groundwater flow contribution using MODFLOW/MT3DMS code instead of COMSOL.
- Angelotti A, Alberti L, La Licata I, Antelmi M (2014) Borehole Heat Exchangers: heat transfer simulation in the presence of a groundwater flow, Journal of Physics: Conference Series 501 012033, doi:10.1088/1742-6596/501/1/012033
Nomenclature table needs to be implemented in the manuscript.
Lines 221-234. could you provide an image that describes all the features shown?
Line 223. Specify the time you use to simulate short-term simulations.
Lines 247-256. Which is the vertical length of the BHE? How is the U-shape pipe reproduced in the COMSOL model? Is there a “BHE” package or is there the need to implement some special boundary conditions? How the cells refinement is? Please give a full description.
Figure 1. Why aren’t the FCS and HCM cases figured out in the image?
Line 278-279. From here, the model will be implemented in 3D way; therefore were previous simulations reproduced in 2D?
Line 280. Why just 6 meters? How do you evaluate this minimal distance?
Line 284. Why do you reduce domain? And, why do you just reduce of 5 and not of 10, at instance?
Line 295. Could you add the true value of Tg in round brackets?
Lines 321-327. Please describe the concepts presented more clearly.
Lines 371-400. The validation is only between numerical model and analytical solutions. This is the first step to have a complete validation in respect to experimental data of a real BHE.
